# Predictive validity of A-level grades and teacher-predicted grades in UK medical school applicants: a retrospective analysis of administrative data in a time of COVID-19

I C McManus [1], Katherine Woolf [1], David Harrison,[1] Paul A Tiffin,[2,3] Lewis W Paton [2], Kevin Yet Fong Cheung,[4] Daniel T Smith [5]

¹Research Department of Medical Education, UCL Medical School, London, UK
²Department of Health Sciences, University of York, York, UK
³Health Professions Education Unit, Hull York Medical School, Hull, UK
⁴Cambridge Assessment, Cambridge, UK
⁵General Medical Council, London, UK

**Correspondence to**
Professor I C McManus;
i.mcmanus@ucl.ac.uk

## ABSTRACT

**Objectives** To compare in UK medical students the predictive validity of attained A-level grades and teacher-predicted A levels for undergraduate and postgraduate outcomes. Teacher-predicted A-level grades are a plausible proxy for the teacher-estimated grades that replaced UK examinations in 2020 as a result of the COVID-19 pandemic. The study also models the likely future consequences for UK medical schools of replacing public A-level examination grades with teacher-predicted grades.

**Design** Longitudinal observational study using UK Medical Education Database data.

**Setting** UK medical education and training.

**Participants** Dataset 1: 81 202 medical school applicants in 2010–2018 with predicted and attained A-level grades. Dataset 2: 22 150 18-year-old medical school applicants in 2010–2014 with predicted and attained A-level grades, of whom 12 600 had medical school assessment outcomes and 1340 had postgraduate outcomes available.

**Outcome measures** Undergraduate and postgraduate medical examination results in relation to attained and teacher-predicted A-level results.

**Results** Dataset 1: teacher-predicted grades were accurate for 48.8% of A levels, overpredicted in 44.7% of cases and underpredicted in 6.5% of cases. Dataset 2: undergraduate and postgraduate outcomes correlated significantly better with attained than with teacher-predicted A-level grades. Modelling suggests that using teacher-estimated grades instead of attained grades will mean that 2020 entrants are more likely to underattain compared with previous years, 13% more gaining the equivalent of the lowest performance decile and 16% fewer reaching the equivalent of the current top decile, with knock-on effects for postgraduate training.

**Conclusions** The replacement of attained A-level examination grades with teacher-estimated grades as a result of the COVID-19 pandemic may result in 2020 medical school entrants having somewhat lower academic performance compared with previous years. Medical schools may need to consider additional teaching for entrants who are struggling or who might need extra support for missed aspects of A-level teaching.

## Strengths and limitations of this study

► This is the first comparison of the predictive validity of teacher-predicted and attained A-level grades for performance in undergraduate and postgraduate assessments 5–8 years later.

► The large sample size of all UK medical applicants from 2010 to 2018 provides adequate statistical power, and the complete population data mean the results are unlikely to be biased.

► The teacher-predicted grades are those provided by schools as a part of university application, and probably form a good proxy for the 'centre-assessment grades', introduced by the Office of Qualifications and Examinations Regulation during the COVID-19 crisis of 2020.

► This study is with medical school applicants only, so that generalisability to students on other university courses is uncertain; however, the overprediction of grades we find in medical school applicants is similar to that found elsewhere for university applicants in general.

## BACKGROUND

… the … exam hall [is] a level playing field for all abilities, races and genders to get the grades they truly worked hard for and in true anonymity (as the examiners marking don't know you). [… Now we] are being given grades based on mere predictions. Yasmin Hussein, letter to *The Guardian*, 29 March 2020[1]

[Let's] be honest, this year group will always be different… Dave Thomson, blog-post on *FFT Educational Lab*[2]

One headmistress commented that 'entrance to university on teachers' estimates may be fraught with unimagined difficulties'. … If there is in the future considerable emphasis on school assessment,

some work of calibration is imperatively called for. James Petch, December 1964.[3]

UK schools closed on 20 March 2020 in response to the COVID-19 pandemic, and key stage 5 (level 3) public examinations such as A levels and Scottish Qualification Authority (SQA) assessments were cancelled for summer 2020 and replaced by a complex system involving teacher assessments of the grades students would have achieved had they taken the examinations. A levels and SQA assessments, like other national examinations in the UK, are normally set and marked anonymously by examination boards which are entirely separate from schools, and teachers usually play no part in this external assessment process. A levels are good predictors of performance at university in general[4] and at medical schools specifically.[5][6] Within this context, the present paper compares achieved A-level grades with teacher-predicted grades, and in particular considers their relative predictive validities for educational outcomes at UK medical schools. The analyses were originally described in May 2020 and published as a preprint[7] while events were still ongoing and outcomes were not known. The present paper maintains much of that structure, and while mostly looking forward from 2020, also in part looks back from the perspective of 2021, meaning that past, present and future tenses are intermingled.

On 3 April 2020, Office of Qualifications and Examinations Regulation (*Ofqual*) in England announced that A level, General Certificate of Secondary Education (GCSE) and other exams under its purview would be replaced by *calculated grades*, at the core of which are teachers' estimates of the grades that their students would attain (called *centre assessment grades* (CAGs)), which would then be moderated by Ofqual using a computer algorithm which included the prior performance of the school attended by candidates (see the Calculated grades subsection for details). The SQA and other national bodies also announced similar processes for their examinations. Inevitably, the announcement of calculated grades resulted in confusion and uncertainty in examination candidates, particularly those needing A levels or SQA Advanced Highers, and therefore they will be available for 2020 applicants; Advanced Highers will not be available and will be estimated) to meet conditional offers for admission to university in autumn 2020. Universities also faced a major problem for student selection, having had A levels taken away, which are 'the single most important bit of information (used in selection)'.[8]

Some of the tensions implicit in calculated grades are well seen in the aforementioned quotation by Yasmin Hussein, a GCSE student in Birmingham, with its clear emphasis that a key strength of current examination systems, such as GCSEs, A levels and similar qualifications, is their *anonymity* and *externality* with assessors who know nothing of the students whose work they are marking. In contrast, the replacement of actual grades attained in the exam hall with what Hussein describes as 'mere predictions' raises a host of questions, not the least being the possibility of bias when judgements are made by teachers.

## Context of the current paper and the situation at the time of writing

Since the appearance of COVID-19 in Europe in early 2020, the situation has been and still is rapidly changing. As mentioned earlier, this paper was originally written in May 2020 but was revised and submitted to the journal, essentially as the preprint but with some additions, in November 2020 when Europe was in the midst of a 'second wave' and England, Wales, Scotland and Northern Ireland, in a second national lockdown. The paper took almost 6 months to be reviewed, with revisions only being requested in May 2021 with the third UK national lockdown still not ended. To help the reader situate the current paper, we explain briefly here what the exam situation was in the UK from April to August 2020, with more details provided in a postscript in section 1 of the online supplemental information.

University selection in the UK for admission in October 2020 began in the autumn, with medical school applicants submitting by 15 October to Universities and Colleges Admissions Service (UCAS) applications for four medical schools. Selection, which may include interviews and other assessments, is usually completed by the end of March, with students being told of offers or rejections. Offers are usually conditional on A levels and other qualifications to be taken in May, with results announced in August. In Spring 2020, as UK universities entered the final phases of the annual academic cycle of student selection, the present paper considered the potential problems of using teacher-estimated grades such as the calculated grades proposed by Ofqual, rather than attained grades obtained in the usual way via examinations. The preprint of May 2020 was circulated primarily for information to medical school admissions tutors. By August 2020, some immediate effects on selection were shown when the algorithms used by regulators resulted in many students, particularly those from historically poorly performing schools, having their expected results adjusted downwards. This forced the Scottish government, followed then by the English and Welsh governments, to accept either teacher-estimated CAGs without moderation by an algorithm, or the calculated grade, whichever was the higher.

As expected in the preprint, given that teacher-estimated grades were found to be higher than attained A-level grades, the scrapping of the algorithm resulted in a significant increase in grades compared with 2019 (https://ffteducationdatalab.org.uk/2020/08/gcse-and-a-level-results-2020-how-grades-have-changed-in-every-subject/), with an immediate impact on the numbers of students meeting university conditional offers. Longer-term impacts are still to be seen, with some likely to result from the lower predictive validity of teacher-estimated

grades, and a likely increase in underperforming students in medical schools and postgraduate training.

## Medical school admissions

This paper mainly concentrates on medical school applications. UK medical education has a range of useful educational measures, including admissions tests during selection, and outcomes at the end of undergraduate training, which are linked together through UK Medical Education Database (UKMED, https://www.ukmed.ac.uk/). UKMED provides a sophisticated platform for assessing predictive validity in multiple entry cohorts in undergraduate and postgraduate training.[9] The current paper should also be read in parallel with a second study from some members of the present team which assesses attitudes and perceptions to calculated grades and other changes in selection of current medical school applicants in the UK Medical Applicants Cohort Study (UKMACS).[10 11]

Fundamental questions about selection in 2020 concerned the likely nature of calculated grades and the extent to which they would predict outcomes to the same extent as currently did *actual or attained grades*. The discussion will involve actual grades, and then four types of teacher-estimated grades: predicted grades (sent to UCAS at application to university), CAGs (submitted by schools to Ofqual in 2020), calculated grades (CAGs adjusted using an algorithm) and forecasted A-level grades (submitted by teachers to exam boards pre-2015 as a quality check for real exam grades). These related but different assessments are summarised in box 1, together with final grades, which were the grades eventually accepted by UCAS and were the higher of the calculated grade or centre assessed grade. It should be noted that we have tried to use 'teacher-predicted' grades only to refer to the grades included as a part of the normal UCAS process, whereas the term teacher-estimated grades is used in a more generic sense.

## Calculated grades

The status of calculated grades was made clear by Ofqual in April 2020:

> The grades awarded to students will have equal status to the grades awarded in other years and should be treated in this way by universities, colleges and employers. On the results slips and certificates, grades will be reported in the same way as in previous years (Ofqual, p6).[12]

The decisions of Ofqual are supported by ministerial statement, and universities and other bodies have little choice therefore but to abide by them, although that does not mean that other factors may not need to be taken into account in some cases, as often occurs when applicants do not attain the grades in conditional offers.

None of the aforementioned means that calculated grades actually *will be* equivalent to conventional attained grades. Calculated grades will not actually *be* attained

---

**Box 1  A-level grades: actual, predicted, centre assessment, calculated, final, forecasted and teacher-estimated grades**

**Actual or attained grades**
The grades awarded by examination boards/awarding organisations based on written and other assessments which are set and marked externally. Typically sat in *May and June of year 13*, with results announced in *mid-August*.

**Predicted grades**
Teacher estimates of the likely attained grades of candidates, provided to UCAS in the *first term of year 13*, and by *15 October* for medical and some other applicants.

**Centre assessment grades**
Used in the production of calculated grades (see further). Provided by examination centres (typically schools) between 1 and 12 June 2020, consisting of teacher-estimated grades and candidate rankings within examination centres.

**Calculated grades**
The final grades to be provided for candidates by exam boards for summer 2020 assessments, in the absence of attained grades. Based on CAGs, with final calculated grades involving standardisation/adjustment by exam boards using an algorithm. Calculated grades 'will have equal status to the grades awarded in other years and should be treated in this way by universities, colleges and employers' (Ofqual). These grades were often referred to as the 'algorithm grades' and were abandoned by the UK government in August 2020.

**Final grades**
The grades used by UCAS in the 2020 admissions cycle – the higher of the teacher estimated grade or the CAG

**Forecasted grades**
Prior to 2015, teachers, in *May of year 13*, provided to exam boards a forecast of the likely grades of candidates along with rankings. Forecasted grades therefore take place later in the academic cycle than predicted grades, close to the time examinations are actually sat.

**Teacher-estimated grades**
Generic term used in this paper to refer to grades estimated by teachers; includes predicted grades, centre assessment grades, calculated grades and forecasted grades.

CAG, centre assessment grade; Ofqual, Office of Qualifications and Examinations Regulation; UCAS, Universities and Colleges Admissions Service.

---

grades; they may well behave differently from attained grades, and in measurement terms they actually *are not* attained grades, even though in administrative and even in legal terms, by fiat, they have to be treated as equivalent. From the perspective of educational research, the key issue is the extent to which calculated grades actually will or can behave in an identical way to attained grades.

In April 2020, Ofqual issued guidance on how calculated grades would be provided for candidates for whom examinations have been cancelled. Essentially, teachers would be required, for individual candidates taking individual subjects within a *candidate assessment centre* (usually a school), to estimate *grades for* candidates, and then to *rank order* candidates within grades, to produce CAGs. A

statistical standardisation process would then be carried out centrally using a computer algorithm. Ranking is needed because standardisation 'will need more granular information than the grade alone'[12] (p.7), presumably to break ties at grade boundaries which occur because of standardisation. Standardisation, to produce calculated grades, would use an algorithm that took into account the typical distribution of results from that centre for that subject in the three previous years, along with aggregated centre data on Standard Assessment Tests (SATS) and previous exam attainment as in GCSEs. (It was this standardisation process that governments reversed in August 2020 after the protests against calculated grades.) This approach is consistent with Ofqual's approach to standard setting. Following Cresswell[13], Of qual has argued that during times of change in assessments, and perhaps more generally, there should be a shift away from 'comparable performance' (ie, criterion-referencing), and that there is an 'ethical imperative' to use 'comparable outcomes' (ie, norm-referencing) to minimise advantages and disadvantages to the first cohort taking a new assessment, as perhaps also for later cohorts as teachers improve at teaching new assessments.[14]

Ofqual said that CAGs, the core of calculated grades, 'are not the same as … predicted grades provided to UCAS in support of university applications',[15] (p.7). Predicted grades in particular are provided by schools in October of year 13 and CAGs in May/June of year 13, 7 months later, when Ofqual says that teachers should also consider classwork, bookwork, assignments, mock exams and previous examinations such as AS levels (taken only by a minority of candidates now) but should *not* include GCSE results or any student work carried out after 20 March. Whether CAGs, or calculated grades—CAGs moderated by the algorithm—will be fundamentally different from predicted grades is ultimately an empirical question, which should be answerable when UCAS data for 2020 are available for medical school applicants in UKMED. In the meantime, and *it is a core and a reasonable assumption*, CAGs and hence calculated grades will probably correlate highly with earlier predicted grades, except for a small proportion of candidates who have improved dramatically from October 2019 to March 2020. Predicted grades, which have been collected for decades, should therefore act as a reasonable proxy in research terms for CAGs and therefore calculated grades, particularly in the absence of any other information.

## Rationale for using A-level grades in selection

Stepping back slightly, it is worth revisiting the reasons that A levels exist and why universities use them in selection. A levels assess at least three things: subject knowledge, intellectual ability and study habits such as conscientiousness.[16] Knowledge and understanding of, say, chemistry are probably necessary for the high-level study of medical science and medicine, to which it provides an underpinning, and experience suggests that students without such knowledge may have problems. A levels also provide evidence for a student's intellectual ability and capability for extended study at a high level. A levels are regarded as a 'gold standard' qualification because of the rigour and objectivity of their setting and marking (see, eg, Ofqual's 'Reliability Programme'[17]). Their measurement is therefore *reliable*, and the presumption is that they are also *valid*, in some of the many senses of that word,[18–20] and as a result are *unbiased*. A crucial assumption is of *predictive validity*, that future outcomes at or after university are higher or better in those who have higher or better A levels, as found in predicting both degree classes in general[4 21 22] and medical school performance in particular.[5 23] There is also an assumption of *incremental validity*, A levels being better predictors than other measures.[6] At the other extreme, A levels could be compared conceptually with, say, a mere assertion by a friend or colleague that 'Oh yes, they know lots of chemistry'. That is likely neither to be reliable, valid nor unbiased, and hence is a base metal compared with the gold standard of A levels. The empirical question therefore is where on the continuum from gold to base metals lie calculated grades or teacher-predicted grades.

The issue of predictive validity has been little discussed in relation to calculated grades, but in a *Times Educational Supplement* survey of teachers, there were comments that 'predictions and staff assessments would never have the same validity as an exam' so that 'Predictions, past assessment data and mock data is not sufficient, and will never beat the real thing in terms of accuracy'.[24] The changes in university selection inevitably meant that difficult policy decisions needed to be made by universities and medical schools. Even in the absence of direct, high-quality, evidence, policy-makers still have an obligation to make decisions, and, therefore it is argued, must take theory, related evidence and so on into account.[25] This paper provides both a review of other evidence and also results on the related issue of predicted grades, which it will be argued are likely to behave in a way that is similar to calculated grades.

## Review of literature on predicted and forecasted grades
### Predicted grades in university selection

A notable feature of UK universities is that selection mostly takes place before A levels or equivalent qualifications have been sat, so offers are largely conditional on later attained grades. As a result, UCAS application forms, since their inception in 1964, have included *predicted grades*, estimates by teachers of the A-level grades a student is likely to achieve. Admissions tutors also use other information in making conditional offers. A majority of applicants in England, applying in year 13 for university entry at age 18, will have taken GCSEs at age 16 in year 11; a few still take AS levels in year 12; some students submit an extended project qualification (EPQ); and UCAS forms also contain candidate statements and school references. Medical school applicants mostly also take admissions tests such as U(K)CAT or Bio-Medical Admissions Test (BMAT) at the beginning of year 13,

and many will take part in interviews or multiple mini-interviews (see https://www.medschools.ac.uk/studying-medicine/making-an-application/entry-requirements).

Predicted grades have always been controversial. A House of Commons Briefing Paper in 2019 noted that the UK was unusual among high-income countries in using predicted grades (https://www.bbc.co.uk/news/education-44525719, and said that

> The use of predicted grades for university admissions has been questioned for a long time. Many critics argue that predicted grades should not be used for university entry because they are not sufficiently accurate and it has been suggested that disadvantaged students in particular lose out under this system.[26] (p.4)

Others have suggested that as well as being 'biased', 'predicting A-level grades is clearly an imprecise science'[27] (p.418). There have been repeated suggestions over the years, none as yet successful, that predicted grades should be replaced with a postqualification application system. As Nick Hillman puts it,

> The oddity of our system is not so much that people apply before receiving their results; the oddity is that huge weight is put on predicted grades, which are notoriously unreliable. … PQA could tackle this… (https://www.hepi.ac.uk/2019/08/14/pqa-just-what-does-it-mean/).

The system of predicted grades is indeed odd, but also odd is the sparsity of academic research into predicted grades. The most important question that seems almost never to have been asked, and certainly not answered, is the fundamental one of whether it is predicted grades or actual grades which are better at predicting outcomes. Petch,[3] in his 1964 monograph, which was one of the first serious discussions of the issues, considers that predicted and actual grades may be fundamentally different, perhaps being 'complementary and not contradictory' (p.29), one being about scholarly attitude and the other about examination prowess, primarily because 'the school knows the candidate as a pupil, knowledge not available to the examiners'. For Petch, either a zero correlation or a perfect correlation between predicted and actual grades would be problematic, the latter perhaps implying that actual grades might be seen as redundant (p.6).

The advent of Ofqual's calculated grades, which are in effect predicted grades carried out by teachers in a slightly different way, means there was a serious need in 2020 to know how effective predicted grades were likely to be as a substitute for attained A-level grades, and the same concern will apply in 2021, with Ofqual implementing a different model for teacher-estimated grades (https://www.gov.uk/government/publications/awarding-qualifications-in-summer-2021/awarding-qualifications-in-summer-2021). Are teacher-predicted grades in fact 'notoriously unreliable', being mere predictions, or do they have equivalent predictive validity as attained grades?

### Research literature on predicted grades

As part of section 1 of the online supplemental information to this paper, we have included a more detailed overview of research studies on predicted grades. Here we will merely provide a brief set of comments.

Most studies look at predictions at the level of individual exam subjects, which at A level are graded from E to A or, from 2010 onwards, from E to A*. The most informative data show all combinations of predicted grades against attained grades, and figure 1 gives an example for medical school applicants. Many commentators, though, look only at overpredictions ('optimistic') and underpredictions ('pessimistic'). Figure 2 summarises data from five studies of university applicants. Accurate predictions occur in 52% of cases when A is the maximum grade and 17% when A* is the maximum grade (and with more categories accuracy is likely to be lower). Grades are mostly overpredicted, in 42% of cases pre-2010 and 73% post-2010, with underprediction rarer at 7% of cases pre-2010% and 10% post-2010. A number of studies have reported that underprediction is more common in lower socioeconomic groups, non-white applicants and applicants from state school or further education.[28–30] A statistical issue means such differences are not easy to interpret, as a student predicted A* cannot be underestimated, and therefore underestimation will inevitably be more frequent in groups with lower overall levels of attainment. This issue is discussed and analysed at length in section 5 of the online supplemental information in relation to applicants from private-sector schools.

Some studies also consider grade-point predictions, the sum of grade scores for the three best attaining subjects, scored A*=12, A=10, B=8, etc. (In some studies a scoring of A*=6, A=5, B=4 is used. The 12, 10, 8 … scoring was introduced so that AS levels, weighted at half an A level, could be scored as A=5, B=4 etc (there being no A* grade at AS-level). For most purposes A*=12, A=10 … is equivalent in all respects to A*=6, A=5, etc, apart from a scaling factor.) In particular, a large study by UCAS[31] showed that applicants 'missing their predictions' (ie, they were overpredicted) tended to have lower predicted grades; lower GCSE attainment; were more likely to have taken physics, chemistry, biology and psychology; and were from disadvantaged areas. To some extent, the same statistical problems of interpretation apply as with analysis at the level of individual exam subjects. For a number of years, UCAS only provided grade-point predictions, and they are included in the P51 data analysed as follows.

### What are predicted grades and how are they made?

UCAS says that 'A predicted grade is the grade of qualification an applicant's school or college believes they're likely to achieve in positive circumstances' (https://www.ucas.com/advisers/managing-applications/predicted-grades-what-you-need-know, accessed 13 April 2020). Later though, the document says predicted grades should be '**in the best interests of applicants** – fulfilment and success at college or university is the end goal' and '**aspirational but**

**A**

| | | Attained Alevel grades | | | | | | |
|---|---|---|---|---|---|---|---|---|
| | | E | D | C | B | A | A* | Total |
| Predicted Alevel grades (points) | E (2 pts) | **200** | 35 | 10 | 5 | 0 | 0 | 255 (0%) |
| | D (4 pts) | 235 | **610** | 155 | 35 | 10 | 0 | 1045 (0%) |
| | C (6 pts) | 635 | 1220 | **2110** | 505 | 95 | 5 | 4570 (2%) |
| | B (8 pts) | 635 | 2095 | 4755 | **7355** | 1695 | 175 | 16715 (7%) |
| | A (10 pts) | 430 | 1925 | 8785 | 35640 | **61950** | 12655 | 121390 (51%) |
| | A* (12 pts) | 50 | 135 | 635 | 6025 | 42815 | **43395** | 93060 (39%) |
| | Total | 2185 | 6020 | 16450 | 49570 | 106570 | 56235 | 237030 |
| | | (1%) | (3%) | (7%) | (21%) | (45%) | (24%) | |

**B**

| | | Attained Alevel grades | | | | | | |
|---|---|---|---|---|---|---|---|---|
| | | E | D | C | B | A | A* | Total |
| Predicted Alevel grades (points) | E (2 pts) | **79%** | 14% | .. | .. | .. | .. | 100% |
| | D (4 pts) | 23% | **58%** | 15% | 3% | .. | .. | 100% |
| | C (6 pts) | 14% | 27% | **46%** | 11% | 2% | .. | 100% |
| | B (8 pts) | 4% | 13% | 28% | **44%** | 10% | 1% | 100% |
| | A (10 pts) | 0% | 2% | 7% | 29% | **51%** | 10% | 100% |
| | A* (12 pts) | 0% | 0% | 1% | 7% | 46% | **47%** | 100% |
| | Total | 1% | 3% | 7% | 21% | 45% | 24% | 100% |

**Figure 1** Predicted versus attained A-level grades for individual subjects in applicants to UK medical schools. Accurate predictions are in bold; yellow indicates overestimates by one grade; orange indicates overestimates by 2+ grades; green denotes underestimates by one grade; blue denotes underestimates by 2+ grades. (A) Counts and (B) attained grades as percentages within predicted grades.

**achievable** – stretching predicted grades are motivational for students, unattainable predicted grades are not' (all emphases in original). Predicted grades should be professional judgements and be data-driven, including the use of 'past Level 2 and Level 3 performance, and/or internal examinations to inform …predictions'.

Few empirical studies have asked how teachers estimate grades, with not much progress since 1964 when Petch said, 'Little seems to be known about measures taken by schools to standardize evaluations of pupils'[3] (p.7). Two important exceptions are the studies of Child and Wilson[32] in 2015 and Gill[33] in May 2018, with only the latter published. Gill sent questionnaires to selected Oxford, Cambridge and Royal Society of Arts Examination Board exam centres concerning chemistry, English literature and psychology exams. Teachers said the most important information used in predicting grades was performance in mock exams, observations of quality of work and commitment, oral presentation, the opinion of other teachers in the same subject and in other subjects, and the head of department. Some teachers raised concerns about the lack of high stakes for mock exams,

which meant that some students did not treat them seriously. AS-level grades were an important aid in making predictions, and there were concerns about the loss of AS levels to help in prediction, as also mentioned elsewhere,[34] and that is relevant to 2020 where most candidates will not have taken AS levels.

Studies considered so far almost entirely are concerned with teacher predictions of A-level grades, since they are important for university admissions. More generally, studies looking at a wider range of teacher estimates, often in younger children, find a tendency for overestimation across a range of skills,[35] with judgements often being systematically lower for marginalised learners.[36] A different position is taken in a genetically informed study of twins, which suggests, in a forcefully worded conclusion, that 'Teachers can reliably and validly monitor students' progress, abilities and inclinations. … For these reasons, we suggest that teacher assessments could replace some, or all, high-stakes exams'.[37] The study, however, uses only correlations as measures of accuracy and cannot assess overestimation or underestimation. Also, teacher ratings were only available at ages 7, 11 and 14, at the same time

| Key: | Blue background: Pre-2010 results | Yellow background: Pre-2000 results | | | | | |
|---|---|---|---|---|---|---|---|
| | Red font: Forecasted grades | Bold, underlined: Averaged results, post 2000 | | | | | |
| | | | | | Under-estimated "Pessimistic" | Accurate | Over-estimated "Optimistic" |
| | Study | Context | Year | A-level range | | | |
| **University applicants overall: A-levels, etc** | | | | | | | |
| | Everett & Papageorgiou (2011) [24] | Predicted Grades | 2009 | A-E | 7% | 52% | 42% |
| | UCAS [27] | Predicted Grades | 2012 | A*-E | 12% | 20% | 68% |
| | Wyness (2016) [25] | Predicted Grades | 2013-15 | A*-E | 9% | 16% | 75% |
| | UCAS [27] | Predicted Grades | 2016 | | 9% | 16% | 74% |
| | UCAS [27] | Predicted Grades | 2017 | | 10% | 16% | 73% |
| | | | | | | | |
| | Petch (1953) [33] | Forecasted Grades | 1940 | School Cert Pass/Fail | 2% | 89% | 9% |
| | Petch (1964) [4] | Non-official forecasted Grades | 1963 | A+B/C+D/E/O/F | 18% | 43% | 39% |
| | Murphy (1979) [34] | Non-official forecasted Grades | 1977 | A-E | 29% | 27% | 44% |
| | Gill and Rushton (2011) [31] | Forecasted Grades | 2009 | A-E | 12% | 55% | 33% |
| | Gill and Chang (2013) [32] | Forecasted Grades | 2012 | A*-E | 13% | 48% | 39% |
| | Gill and Benton (2015) [30] | Forecasted Grades | 2014 | A*-E | 14% | 43% | 43% |
| | Gill (2019) [28] | Non-official forecasted Grades | 2018 | A*-E | 20% | 45% | 35% |
| | | | | | | | |
| | | **Mean Predicted Grades** | Pre-2010 | A-E | **7%** | **52%** | **42%** |
| | | **Mean Forecasted Grades** | Pre-2010 | A-E | **20%** | **42%** | **39%** |
| | | **Mean Predicted Grades** | Post-2010 | A*-E | **10%** | **17%** | **73%** |
| | | **Mean Forecasted Grades** | Post-2010 | A*-E | **15%** | **46%** | **39%** |
| | | | | | | | |
| **Medical school applicants and students: Alevels and other qualifications** | | | | | | | |
| | Students: Lumb & Vail [38] | Predicted grades | 1995 | A-E | 7% | 52% | 41% |
| | Applicants: Alevels (this study) | Predicted grades | 2010-18 | A*-E | 7% | 49% | 45% |
| | Applicants: EPQ (this study) | Predicted grades | 2010-19 | A*-E | 14% | 52% | 34% |
| | Applicants: SQA Adv. Highers (this study) | Predicted grades | 2010-18 | A-D | 3% | 60% | 38% |
| | | **Mean Predicted Grades (Medics)** | | | **8%** | **53%** | **39%** |
| | | | | | | | |
| **GCSE grades: All candidates** | | | | | | | |
| | Gill and Chang (2015) [36] | Forecasted GCSE Grades | 2013 | A*-G,U | 12% | 47% | 41% |
| | Gill and Benton (2015) [35] | Forecasted GCSE Grades | 2014 | A*-G,U | 14% | 44% | 42% |
| | | **Mean Forecasted GCSE grades** | | | **13%** | **45%** | **42%** |

**Figure 2** Overestimated, underestimated and accurate predicted grades in various studies. Black font: predicted grades; red font: forecasted grades; yellow background: pre-2000; blue background: pre-2010; bold, underlined: averaged results post-2000.

as standardised tests are carried out, but were not available for GCSEs at age 16, or for A levels and university entrance at age 18, and as such are not informative for the purposes of the present study.

### Predicted grades in other key stage 5 qualifications than A levels

Almost all studies on predicted grades have considered A levels, with a few occasional exceptions looking at GCSEs. We know of no studies on the EPQ in England, of Scottish Highers and Advanced Highers, or any other qualifications. Section 3 of the online supplemental information includes data on both EPQ and SQA examinations.

### Forecasted grades

Until 2015, teachers in the May of school year 13 provided awarding organisations with *forecasted grades*, and those forecasts in part contributed to quality control of grades by the boards. Since forecasted grades were produced 5 to 7 months after predicted grades, and closer to the exam date, they might be expected to be more accurate than predicted grades, being based on better and more recent information. Forecasted grades are important as they are more similar than predicted grades to the proposed calculated grades in the way they are calculated, and it is noted that 'they may differ somewhat from the predicted grades sent to UCAS as part of the university application process'.[38] Three formal analyses are available, for candidates in 2009,[39] 2012[40] and 2014,[38] and four other studies from 1940,[41] 1963,[3] 1977[42] and 2018[33] are also available, with one post-2000 study before A* grades

were introduced and three after (figure 2). Petch[41] also provides a very early description of forecasted grades, looking at teachers' predictions of pass or fail in school certificate examinations in 1940, which also show clear overprediction.

Forecasted A-level grades are similar in accuracy to predicted grades pre-2010 (42% vs 52%) but are less accurate post-2010 (47% vs 17%), in part due to a drop in accuracy of predicted grades when A* grades are available. Despite there being *no aspirational or motivational reasons for teachers to overpredict forecasted grades*, particularly in the 1977 and 2018 studies, overprediction nevertheless remains as frequent as with predicted grades (pre-2010: 39%, post-2010: 37%) and remains more common than underprediction (pre-2010: 20%, post-2010 16%). Overall, it is perhaps possible that calculated grades may be somewhat more accurate than predicted grades, but forecasted grades appear broadly in their behaviour to predicted grades. Two sets of forecasted grades are available for GCSEs,[43 44] and they show similar proportions of overprediction and underprediction as do results for A levels. Overprediction seems to be a feature of all predictions by teachers.

The three non-official studies of forecasted grades also asked teachers to rank-order candidates, a procedure which was included in calculated grades. The 1963 data[3] found a median correlation of rankings and exam marks within schools of 0.78, the 1977 data[42] a correlation of 0.66[42] and the recent 2018 data[33] a correlation of about

0.82. The three estimates (mean r=0.75) are somewhat higher than a meta-analytic estimate of 0.63 (SE=0.03) for teachers' ability to predict academic achievement.[45]

The Gill study[33] is also of interest as one teacher commented on the difficulty of providing rankings with 260 students sitting one exam, and the author noted that 'it was easier for smaller centres to make predictions because they know individual students better' (p.42), with it also being the case that responses to the questionnaire were more likely to come from smaller centres. The 1963 study of Petch,[3] as well as commenting on 'considerable divergencies … in the methods by which estimates were produced' (p.27), as in the variable emphasis put on mock exams, also adds that 'some of the comments from schools suggested that at times there may be a moral ingredient lurking about some of the estimates' (p.28).

Overall, it seems possible but unlikely that calculated grades might be more accurate than predicted grades, but they also make clear the problems shown by teachers in ranking and grading candidates. It also remains possible that examining boards have far more extensive and unpublished data on forecasted grades that they intend to use in assessing the likely effectiveness of calculated grades.

### Applicants to medical school

So far, this review section has been entirely about university applicants across all subjects and the entire range of A-level grades. Only a handful of studies have looked at predicted grades in medical school applicants.

Lumb and Vail emphasised the importance of teacher-predicted grades since they determine in large part how shortlisting takes place.[46] In a study of 1995 applicants, they found 52% of predictions were accurate; 41% were overestimated; and 7% were underestimated,[46] values very similar to those reported in university selection in general (figure 2).

A study by one of the present teams used path modelling to assess the causal inter-relationships of GCSE grades, predicted grades, receipt of an offer, attained A-level grades and acceptance at medical school.[47] Predicted grades were related to GCSE grades (beta=0.89), and attained A-level grades were predicted by both GCSE grades (beta=0.44) and predicted A-level grades (beta=0.74). The study supports claims that teachers may well be using GCSE grades in part to provide predicted grades, which is perhaps not unreasonable, given the clear correlation.

Richardson et al,[48] in an important and seemingly unique study, looked at the relative predictive validity of predicted as compared with attained A-level grades. Using a composite outcome of preclinical performance, they found that there was a minimal correlation with predicted grades (r=0.024) compared with a correlation of 0.318 (p<0.001) with attained A-level grades. To our knowledge, this is the only study of any sort assessing the predictive validity of predicted versus attained A-level grades.

### Present study

Although calculated grades are novel and untested in their details, predicted grades have been around for half a century, and there is also a small literature on forecasted grades. This paper will try to answer several empirical questions about predicted grades, for which data are now available in UKMED. Predicted grades will then be used, faute de mieux, to make inferences about the likely consequence of using calculated grades.

### Empirical questions to be addressed
#### *Relationship between predicted and attained grades in medical school applicants*

Few previous studies have looked in detail at this high-performing group of students. We will also provide brief results on Scottish Highers and Advanced Highers, and the EPQ, neither of which has been discussed elsewhere to our knowledge.

#### *Predictive validity of predicted grades in comparison with attained grades*

A fundamental question concerning calculated grades is whether teacher-predicted grades are better or worse at predicting outcomes than are actual A-level grades. The relationship between predicted grades and actual grades cannot itself answer that question. Instead, what matters is the relative performance of predicted and actual grades in predicting subsequent outcomes at the end of undergraduate or postgraduate training. The only relatively small study on this of which we are aware in medical students[48] found that only actual grades had predictive validity.

### METHOD

The method provided here is brief. A fuller description including a detailed table of measures can be found in section 2 of the online supplemental information. Overall, the project is *UKMEDP112*, approved by the UKMED Research Group in May 2020, with data coming from two separate but related UKMED projects, both of which included predicted grades.

Project *UKMEDP089*, 'The UK Medical Applicant Cohort Study: Applications and Outcomes Study', approved on 7 December 2018, with Professor Katherine Woolf as principal investigator, is an ongoing analysis of medical student selection as a part of UKMACS (https://ukmacs.wordpress.com/). The data upload of 21 January 2020 included detailed information from UCAS and Higher Education Statistics Agency Limited (HESA) on applicants for medicine from 2007 to 2018.

Project *UKMEDP051*, 'A comparison of the properties of BMAT, GAMSAT and UKCAT', approved on 25 September 2017, with Professor Paul Tiffin as principal investigator, is an ongoing analysis of the predictive validity of admissions tests and other selection methods such as A levels and GCSEs in relation to undergraduate and postgraduate attainment. The present analysis

used the download files dated 13 May 2019 (UKCAT51_ APP_ALL_DATA_13052019_FILE1.SAV and UKCAT51_ APP_ALL_DATA_13052019_FILE2.SAV). UCAS data are included, although when the present analysis began, the file had not yet included the detailed subject-level information available in UKMEDP089. (An upload for P51 was made available on 20 April 2020 but was not included in the present analyses.) Outcome data for the P51 dataset are extensive, and in particular undergraduate progression data are included, such as UKFPO Educational Performance Measure (EPM) and Situational Judgement Test (SJT) and Prescribing Safety Assessment (PSA), as well as performance on some postgraduate examinations (Membership of the Royal Colleges of Physicians (MRCP) part 1 and Membership of the Royal College of Surgeons (MRCS) part A).

Data from HESA and hence UKMED are required to be reported using their rounding and suppression criteria (https://www.hesa.ac.uk/about/regulation/data-protection/rounding-and-suppression-anonymise-statistics), and those criteria have been used for all UKMED data. In particular, the presence of a zero or the absence of a percentage may not always mean that there are no individuals in a cell of a table, and all integers are rounded to the nearest 5.

## RESULTS

A fuller description of the results can be found in section 3 of the online supplemental information.

### Relationships between predicted and actual grades in medical school applicants

#### Predicted and actual A-level grades for individual A-level examinations

Figure 1 shows the relationship between predicted and attained A-level grades for 237 030 examinations from 2010 to 2018 (ie, assessments including A* outcomes). Of predicted grades, 39.3% are A* compared with 23.7% of attained grades. Figure 1A shows predicted grades in relation to attained grades, with bold font for accurate predictions, green and blue shading for underprediction, and orange and red shading for overprediction. Overall, 48.8% of predicted grades are accurate, which is higher than for university applications in general (see figure 2), reflecting the high proportion of A and A* grades (69%). Overprediction occurred in 44.7% of cases, and underprediction occurred in 6.5% of cases. Figure 1B shows the data as percentages. About a half of A* predictions result in an attained A grade, and over a third of predicted A grades result in grade B or lower. Predicted and attained grades have a Pearson correlation of r=0.63.

#### Differences between A-level subjects

There is little in the literature on the extent to which different A-level subjects may differ in the accuracy of their predictions, perhaps with different degrees of bias or correlation. Detailed results are presented in section 3

of the online supplemental information. Overall, biology, chemistry, maths and physics are very similar in terms of overprediction and correlation with actual grades. However, general studies is particularly overestimated compared with other subjects.

### EPQ and SQA Advanced Highers

Section 3 of the online supplemental information contains information on these qualifications. SQA Advanced Highers, as well as the EPQ, show similar proportions of overestimation as other qualifications (see figure 2).

### Reliability of predicted and attained A-level grades

Considering the best three A-level grades, the reliability of an overall score can be calculated from the correlations of the individual subjects. For 66 006 candidates with at least three paired predicted and actual grades, Cronbach's alpha was 0.827 for actual grades and 0.786 for predicted grades, with a highly significant difference. The difference may in part reflect the higher proportion of A* grades in predicted than actual grades, and hence a greater ceiling effect, but may also reflect greater measurement precision in the marking of actual A levels.

#### How reliable are attained A-level grades?

Attained A-level grades, like any behavioural measurement, are not perfectly reliable, in the sense that if a candidate took a parallel test containing equivalent but different items, it is highly unlikely that they would get exactly the same mark as on the first attempt. They may, for instance, have been lucky (or unlucky) at their first attempt, being asked questions on topics which they happened to have studied or revised more (or revised less), and so on. Reliability is a technical subject (see https://www.gov.uk/government/publications/reliability-of-assessment-compendium for a range of important papers commissioned and published by Ofqual) with many different approaches.[49 50] For continuous measures of raw scores, the reliability can be expressed as a coefficient such as alpha (and in one A-level math test in 2011, alpha for the full test was about 0.97,[51] although it is suggested that value is unusually high). Boards though do not report raw scores but instead award grades on a scale such as A* to E. The 'classification accuracy' of grades is harder to estimate and is greater with fewer grade points, wider grade intervals and a wide spread of candidate ability.[51] There seem to be few published estimates of classification accuracy for A levels, although they do exist for GCSEs and AS-levels.[51]

Estimating classification accuracy for the present high-attaining group of medical school applicants is not easy. A fundamental limit for any applicant is that predicted grades cannot possibly predict actual grades better than attained grades predict themselves (the reliability or classification accuracy). However, from considering the correlation of the three best predicted and actual grades, it is unlikely that such a limit has currently been reached. The correlation of actual with predicted grades is 0.585,

and the alpha reliabilities of 0.827 for actual grades and 0.786 for predicted grades (see previous discussion). The disattenuated correlation between predicted and actual grades is therefore $0.585/(\sqrt{(0.827\times0.786)})=0.726$, which is substantially less than 1, with predicted grades accounting for only about a half of the true variance present in actual grades. If the disattenuated correlation were close to 1, then it could be argued that predicted grades were doing as well as they could possibly do, given that attained grades are not perfectly reliable, but that is clearly far from the case.

### True scores and actual scores

From a theoretical, psychometric point of view, it could be argued that it is neither actual nor predicted grades which need to be estimated for applicants, but their 'true ability scores', or the 'latent scores', to use the technical expressions, of which predicted and actual grades are but imperfect estimates. In an ideal world, that would be the case, and a well-constructed exam tries to get as close as possible to true scores. However, it is not possible to know true scores (and if it were the boards would provide selectors with those scores). Selection itself does not work on true scores but on the actual grades that are written down by teachers for predicted grades and as grades on exam result certificates by boards. They are the currency in which transactions are conducted during selection, so that a predicted grade of less than a certain level means a candidate will not get a conditional offer, and likewise too low an actual grade means a candidate holding a conditional offer will be rejected. For that reason, it is not strictly the correlation of predicted and actual grades which matters, the two measures being treated as symmetric, but the forward prediction of actual grades from predicted grades, that is, the actual grades conditional on the predicted grades (as shown in figure 1B).

### Predictive validity of predicted and attained A-level grades in medical students

#### Predictive validity in UKMEDP051

The version of the P51 data used here consists entirely of applicants applying to medical schools, but there is also follow-up into undergraduate and postgraduate training. Predicted A-level grades were available only for the UCAS application cycles of 2010–2014 (ie, applying for university entry in October 2009, for the academic year 2010/11, etc) and consisted of a single score in the range 4–36 points, based on the sum of the three highest predicted grades, scored as A*=12, A=10, etc. The modal score for 38965 applicants was 30 (equivalent to AAA; mean=31.17; SD=3.58; median=32; 5th, 25th, 75th and 95th percentiles=26, 30, 34 and 36). For simplicity, the study was restricted to applicants aged 18 in the year of application who had both predicted and attained A levels, which also ensured the sample contained only first applications for non-graduate courses, from candidates who had not taken pre-2010 A-levels, when A* grades were not available. Overall, 22955 applicants were studied. Other selection measures included were GCSEs (mean grade for best eight grades), as well as U(K)CAT and BMAT scores, based on the most recent attempt which for cases was also the first attempt. For simplicity, we used the total of the four subscores of U(K)CAT, and the total of section 1 and 2 scores for BMAT.

Follow-up is complicated as application cohorts enter medical school in different years and spread out in time through medical school and training. Figure 3 uses an Ibry chart[52–55] to show the educational progression of typical 18-year-old medical school entrants, through to postgraduate qualifications. There are, however, many variants on this theme. The horizontal axis shows academic years (September–August) and training years (August–July),

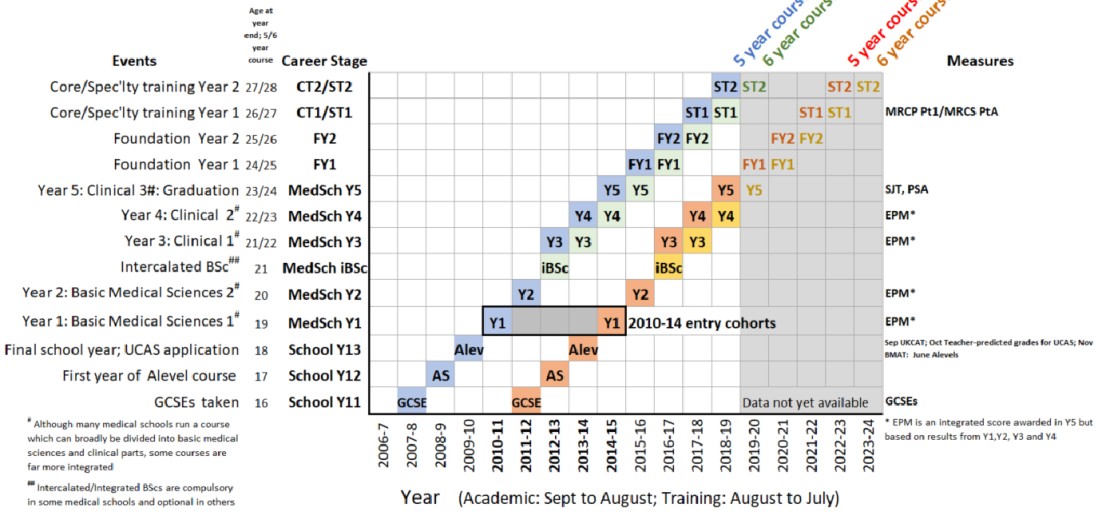

**Figure 3** An Ibry chart illustrating the progression of the 2010–2014 medical school entry cohorts through secondary schooling, application to medical school, undergraduate and postgraduate training, with the timing of key events shown. See text for further details. ALEV, A level; EPM, Educational Performance Measure; MRCS, Membership of the Royal College of Surgeons; PSA, Prescribing Safety Assessment; SJT, Situational Judgement Test.

with career stages, key events and measures used on the vertical axis, with coloured boxes indicating typical students, although there are many variants on entry and progression. The blue boxes show typical students on a 5-year course who entered medical school in October 2010 at the age of 18. They would have taken GCSEs in June 2008 in school year 11, in the 2007/2008 academic year, and some would have taken AS levels in June 2009. Applicants would have taken aptitude tests in school year 13, most taking either U(K)CAT or BMAT but some taking both tests. U(K)CAT would have been taken between July and September 2009 and BMAT in November 2009. UCAS applications are submitted in October, with teachers providing teacher-estimated grades. Note that U(K)CAT results are known before UCAS applications, but BMAT results are not known until after application. A levels would have been taken in May–June 2010, with results known in August 2010, and successful applicants entering medical school in October 2010. Students on a 5-year course would start the second medical school year in October 2011, the third and fourth years in 2012 and 2013, and during their final year beginning in October 2014, they would take the SJT and PSA tests and be awarded an EPM score, with graduation in May 2015. The first of the two foundation years starts in August 2015, and core or specialist training begins in August 2017. Medical students at some schools take an optional or a compulsory intercalated BSc (iBSc) between years 2 and 3. As a result, they are then a year later in progressing to the later stages and are shown by the green boxes in figure 3. Although years are broadly divided into basic medical science and clinical stages, some medical schools have courses which are far more integrated.[56]

The aforementioned description is for 18 year olds entering the 2010 entry cohort. The present study included the 2010–2014 entry cohorts (shown by the solid black box in the lower left of figure 3). For simplicity, the last of those cohorts is the only other one, the 2014 entrants having red boxes to show progression for a 5-year course and orange for a 6-year course including an iBSc. It should be re-emphasised that all career trajectories are idealised, and in reality, students and doctors have many and varied training trajectories.

Data were available up until the 2018 academic year, and years after that are therefore shown greyed out in figure 3. Although all cohorts had data for EPM, SJT and PSA, the later entry cohorts are less likely to have post-graduate qualifications.

Undergraduate outcome measures were for simplicity restricted to the deciles of the UKFPO's EPM, the raw score of the UKFPO's SJT and the score relative to the pass mark of the PSA, all at first attempt. Relatively few doctors, mostly from the earlier cohorts, had progressed through to postgraduate assessments, but sufficient numbers for analysis were present for MRCP (UK) part 1 and MRCS part A, with scores being analysed at the first attempt. It should be noted that while U(K)CAT, BMAT, PSA, SJT and postgraduate assessments are *nationally*

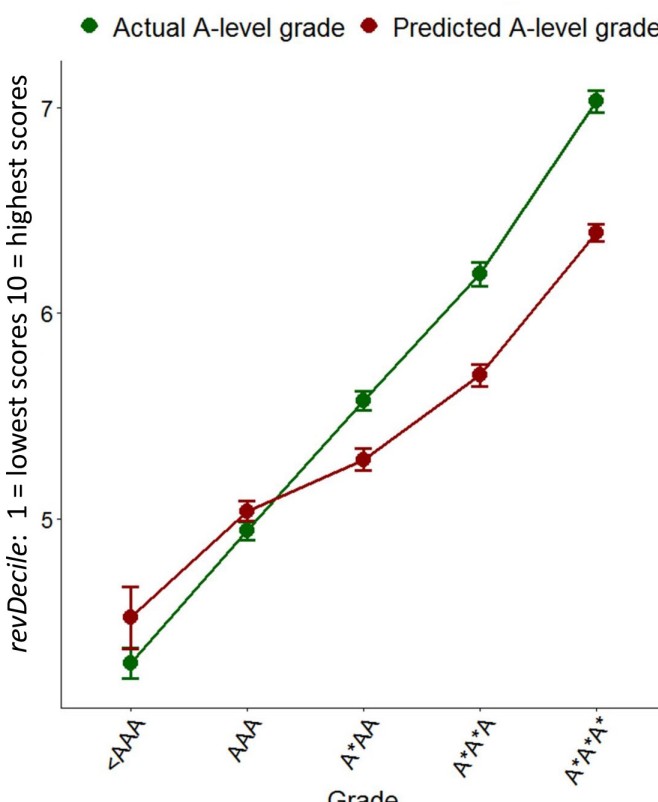

**Figure 4** Mean Educational Performance Measure revDeciles (95% CI) in relation to actual A-level grades (green) and predicted A-level grades (red).

*standardised*, EPM deciles are *locally standardised* within medical schools.

EPM is a complicated measure summarising academic progression through the first 4 years of medical school, with individual medical schools deciding what measures to include,[57] and expressed as deciles *within* each school and graduating cohort year. EPM is used here as the main undergraduate outcome measure. EPM deciles are confusing, as UKFPO scores them in the reverse of the conventional order, the 1st decile being highest performance and the 10th the lowest (https://foundationpro-gramme.nhs.uk/wp-content/uploads/sites/2/2019/11/UKFP-2020-EPM-Framework-Final-1.pdf). Here, for ease of interpretation, we reverse the scoring in what we call *revDecile*, so that higher scores indicate higher performance. It should also be remembered that deciles are not an equal interval scale (figure 4).

Correlations between the measures are summarised in figure 5. Large differences in Ns reflect some measures being used in applicants during *selection* and others being outcome measures that are only present in *entrants*, as well as the smaller numbers of doctors who had progressed to postgraduate assessments. The distinction is emphasised by dividing the correlation matrix into three separate parts. Correlations of selection and outcome measures necessarily show *range restriction* because candidates have been selected on the basis of the selection measures, and

| | | | Selection measures applicants | | | | | Undergraduate outcome measures | | | Postgraduate outcome measures | |
|---|---|---|---|---|---|---|---|---|---|---|---|---|
| | | | GCSE grades | Predicted Alevels | Alevel grades | UKCAT | BMAT | EPM | SJT | PSA | MRCP(UK) Part 1 | MRCS Part A |
| **Selection measures in all applicants** | **GCSE grades** | r | 1 | 0.452 | 0.421 | 0.265 | 0.223 | 0.180 | 0.190 | 0.201 | 0.212 | 0.173 |
| | | N | | 22150 | 22150 | 22145 | 4935 | 12230 | 12185 | 12265 | 890 | 430 |
| | **Predicted A-level grades** | r | 0.452 | 1 | 0.585 | 0.272 | 0.326 | 0.198 | 0.160 | 0.226 | 0.283 | 0.181 |
| | | N | 22150 | | 22955 | 22520 | 5225 | 12560 | 12515 | 12600 | 910 | 440 |
| | **Attained A-level grades** | r | 0.421 | 0.585 | 1 | 0.326 | 0.416 | 0.297 | 0.195 | 0.306 | 0.421 | 0.358 |
| | | N | 22150 | 22955 | | 22520 | 5225 | 12560 | 12515 | 12600 | 910 | 440 |
| | **UKCAT total** | r | 0.265 | 0.272 | 0.243 | 1 | 0.483 | 0.115 | 0.243 | 0.238 | 0.200 | 0.181 |
| | | N | 22145 | 22520 | 22520 | | 5080 | 12385 | 12340 | 12420 | 900 | 435 |
| | **BMAT sections 1 and 2** | r | 0.223 | 0.326 | 0.416 | 0.483 | 1 | 0.089 | 0.239 | 0.321 | 0.378 | 0.319 |
| | | N | 4935 | 5225 | 5225 | 5080 | | 4850 | 4840 | 4875 | 450 | 240 |
| **Undergraduate outcome measures** | **UKFPO EPM decile** | r | 0.180 | 0.198 | 0.297 | 0.115 | 0.089 | 1 | 0.319 | 0.470 | 0.509 | 0.535 |
| | | rTPa | 0.213 | 0.251 | 0.403 | 0.149 | 0.101 | - | - | - | - | - |
| | | N | 12230 | 12560 | 12560 | 12385 | 4850 | | 12515 | 12505 | 905 | 440 |
| | **UKFPO SJT score** | r | 0.190 | 0.160 | 0.195 | 0.243 | 0.239 | 0.319 | 1 | 0.346 | 0.351 | 0.274 |
| | | rTPa | 0.223 | 0.203 | 0.267 | 0.310 | 0.267 | - | - | - | - | - |
| | | N | 12185 | 12515 | 12515 | 12340 | 4840 | 12515 | | 12475 | 905 | 435 |
| | **PSA score** | r | 0.201 | 0.226 | 0.306 | 0.238 | 0.321 | 0.470 | 0.346 | 1 | 0.500 | 0.483 |
| | | rTPa | 0.236 | 0.287 | 0.415 | 0.305 | 0.360 | - | - | - | - | - |
| | | N | 12265 | 12600 | 12600 | 12420 | 4875 | 12505 | 12475 | | 910 | 440 |
| **Postgraduate outcome measures** | **MRCP(UK) Part 1** | r | 0.212 | 0.283 | 0.421 | 0.200 | 0.378 | 0.509 | 0.351 | 0.500 | 1 | ... |
| | | rTPa | 0.272 | 0.360 | 0.601 | 0.273 | 0.398 | 0.586 | 0.391 | 0.576 | - | - |
| | | N | 890 | 910 | 910 | 900 | 450 | 905 | 905 | 910 | | 10 |
| | **MRCS Part A** | r | 0.173 | 0.181 | 0.358 | 0.181 | 0.319 | 0.535 | 0.274 | 0.483 | ... | 1 |
| | | rTPa | 0.196 | 0.216 | 0.519 | 0.282 | 0.313 | 0.618 | 0.306 | 0.575 | - | - |
| | | N | 430 | 440 | 440 | 435 | 240 | 440 | 435 | 440 | 10 | |
| **All outcome measures** (unweighted mean) | **Undergraduate (n=3)** | r | 0.190 | 0.195 | 0.266 | 0.199 | 0.216 | - | - | - | - | - |
| | | rTPa | 0.224 | 0.247 | 0.362 | 0.255 | 0.243 | - | - | - | - | - |
| | **Postgraduate (n=2)** | r | 0.193 | 0.232 | 0.390 | 0.191 | 0.349 | 0.522 | 0.313 | 0.492 | - | - |
| | | rTPa | 0.234 | 0.288 | 0.560 | 0.278 | 0.356 | 0.602 | 0.349 | 0.576 | - | - |
| | **Undergraduate and Postgraduate (n=5)** | r | 0.191 | 0.210 | 0.315 | 0.195 | 0.269 | - | - | - | - | - |
| | | rTPa | 0.228 | 0.263 | 0.441 | 0.264 | 0.288 | - | - | - | - | - |
| **Range restriction (uX) = SD(entrants)/SD(applicants)** | **EPM, SJT & PSA** | uX | 0.955 | 0.958 | 0.888 | 0.890 | 0.994 | - | - | - | - | - |
| | **MRCP(UK) Pt1** | uX | 0.833 | 0.954 | 0.883 | 0.842 | 1.055 | 0.962 | 0.997 | 0.957 | - | - |
| | **MRCS Part A** | uX | 0.985 | 0.998 | 0.835 | 0.761 | 1.123 | 0.946 | 0.958 | 0.927 | - | - |

**Figure 5** Correlation matrix of selection measures, undergraduate outcome measures and postgraduate outcome measures (separated by grey lines for clarity). Cells indicate simple Pearson correlations (R, in blue), construct-level predictive validity (rtPA, in red) and sample size (N, in black). EPM, Educational Performance Measure; MRCP, Membership of the Royal Colleges of Physicians; MRCS, Membership of the Royal College of Surgeons; PSA, Prescribing Safety Assessment; SJT, Situational Judgement Test.

likewise doctors taking postgraduate examinations may be self-selected for earlier examination performance.

Figure 5 contains much of interest (see also section 3 of the online supplemental information), but the most important question for present purposes is the extent to which predicted and attained A-level grades (shown in pink and green in figure 5) differ in their prediction of the five outcome measures, remembering that undergraduate outcomes are typically 5 or 6 years after selection, and postgraduate outcomes are 7 or 8 years after selection.

Attained A levels predict EPM with a simple Pearson correlation of r=0.297 compared with a correlation of only 0.198 for predicted grades (simple correlations, *r*, are shown in blue in figure 5). N is large for these correlations and hence the difference, using a test for correlated correlations[58] is highly significant (Z=12.6, p<$10^{-33}$). Multiple regression (see section 3 of the online supplemental information) suggests that predicted grades may have a small amount of predictive variance which is not shared with attained A levels. Figure 4 shows mean EPM revDecile scores in relation to actual and predicted A levels. The slope of the line is clearly less for predicted

A levels, showing a less good prediction. It is also clear that attained grades predict well, with A*A*A* entrants scoring an average of two deciles higher at the end of the course than those with AAA grades, each extra grade raising average performance by about two-thirds of a decile. In contrast, the slope is less for predicted grades, being slightly less than half a decile per predicted A-level grade. The broad pattern of results is similar for the other undergraduate outcomes, SJT and PSA, and is shown in section 3 of the online supplemental information.

The two postgraduate outcome measures, MRCP (UK) examination part 1 and MRCS part A, although both based on smaller but still substantial numbers of doctors, are still significant, with actual grades correlating more highly with MRCP (UK) part 1 (r=0.421) than do predicted grades (r=0.283; Z=4.54, p=0.000055). Likewise, actual grades correlate more highly with MRCS part A (r=0.421) than do predicted grades (r=0.358; Z=3.67, p=0.000238).

The simple correlations (*r*) in figure 5 are inevitably range restricted as A-level grades and predicted A-level grades have themselves been used as a part of the selection process. Taking range restriction into account using

the method of Hunter *et al*[6 59] (see also Fife *et al*[60]), who used $u_X$, the ratio of SD in the predictors in the unrestricted and the restricted population, with values below 1 indicating more range restriction. Figure 5 shows $u_X$ (uX) at the bottom of the columns, and it can be seen that it is much lower for actual A-level grades than predicted A-level grades, suggesting that actual grades are more important in the selection process than are predicted grades. Construct-level predictive validity (CLPV)[6] can be calculated, taking reliability of measures into account, using 0.827 for attained A levels and 0.785 for predicted A levels (see earlier), with all other reliabilities set at 0.9 in the absence of better estimates. Note that the calculation, unlike that carried out previously,[6] for simplicity does not take censorship/ceiling effects of A levels into account, and a fuller analysis will be presented elsewhere. The CLPV, $\rho_{TPa}$ (shown as rTPa in figure 5), given the greater range restriction, is relatively higher for actual A-level grades than for predicted A-level grades. CLPV for predicting EPM is 0.403 for actual A-level grades compared with 0.251 for predicted A-level grades. For predicting postgraduate qualifications, CLPV for MRCP (UK) part 1 and MRCS part A are 0.601 and 0.519 for attained A-level grades compared with 0.360 and 0.216, respectively, for predicted A-level grades.

There are suggestions that predicted grades may not be equivalent in candidates from state schools and private schools, with grades being predicted more accurately in independent schools.[28 29] That is looked at in section 5 of the online supplemental information, and while there is clear evidence, as found before in the UKCAT-12 study,[61] that private school entrants underperform relative to expectations based on their A levels, there is no evidence that predicted grades behave differently in candidates from private schools.

A practical question relevant to calculated grades concerns the extent to which, in the absence of attained A-level grades, other selection measures such as GCSEs, U(K)CAT and BMAT can replace the predictive variance of attained A-level grades. That will be considered for EPM where the sample sizes are large. Attained grades alone give r=0.297, and predicted grades alone give r=0.198, accounting for less than half as much outcome variance. Adding GCSEs to a regression model including just predicted grades increases multiple R to 0.225, and also including U(K)CAT and BMAT increases it to 0.231, which though is still substantially less than the 0.297 for attained A-levels alone. In the absence of attained A-level grades, prediction is improved by including GCSEs and U(K)CAT or BMAT, but the prediction still falls short of that for actual A levels alone.

### Modelling the effect of only predicted grades being available for selection

In the context of the 2020 pandemic, an important question is the extent to which future outcomes may change as a result of selection being in terms of calculated grades. Calculated grades themselves were not known at the time of the study, but predicted grades are probably a reasonable surrogate for them in the first instance. A modelling exercise was therefore carried out whereby the numbers of students in the various EPM revDeciles were tabulated in relation to predicted grades at five grade levels, 36 pts≡A*A*A*, 34 pts≡A*A*A, 32 pts≡A*AA, 30 pts≡AAA and ≤28 pts≡≤AAB, with the probability of each decile found for each predicted A-level band. Assuming that selection results in the usual numbers of entrants with grades of A*A*A*, A*A*A, etc, but based on calculated grades rather than actual grades, the expected numbers of students in the various EPM deciles can be found. Figure 6 shows deciles as standard UKFPO deciles (1=highest), UKFPO scores (43=highest) and revDeciles (10=highest). The blue column shows the actual proportions in the deciles based on attained A-level grades. Note that for various reasons, there are not exactly equal proportions in the 10 deciles. (In part, this reflects the fact that some students, particularly weak ones, are given an EPM score, but then fail finals.) Based on selection on attained A-level grades, there are 7.2% of students in the lowest-performing decile, compared with an expected

| | | UKFPO | | Selection grades: | | | Absolute | Relative |
|---|---|---|---|---|---|---|---|---|
| | Decile | score | *RevDecile* | Attained | Predicted | Odds Ratio | difference | increase |
| Worst | 10 | 34 | 1 | 7.2% | 8.1% | 1.141 | 0.9% | 13.0% |
| | 9 | 35 | 2 | 9.4% | 10.6% | 1.135 | 1.1% | 12.0% |
| | 8 | 36 | 3 | 10.1% | 11.1% | 1.107 | 1.0% | 9.5% |
| | 7 | 37 | 4 | 10.7% | 11.2% | 1.052 | 0.5% | 4.6% |
| | 6 | 38 | 5 | 10.7% | 10.8% | 1.003 | 0.0% | 0.3% |
| | 5 | 39 | 6 | 10.6% | 10.4% | 0.978 | -0.2% | -2.0% |
| | 4 | 40 | 7 | 10.7% | 10.4% | 0.970 | -0.3% | -2.7% |
| | 3 | 41 | 8 | 10.3% | 9.7% | 0.935 | -0.6% | -5.8% |
| | 2 | 42 | 9 | 10.2% | 9.1% | 0.882 | -1.1% | -10.7% |
| Best | 1 | 43 | 10 | 10.1% | 8.8% | 0.853 | -1.4% | -13.4% |

**Figure 6** Predicted decile outcomes if selection were on predicted A-level grades (blue) rather than actual A-level grades (orange).

proportion of 8.1% for selection on predicted grades, an increase of 0.9% percentage points, which is a relative increase of 13.0% in the proportion of the lowest decile, with an OR of 1.141 of attaining the lowest decile. For the highest-scoring decile, the proportion decreases from 10.1% with actual A-level grades to 8.8% if predicted A-level grades are used, an absolute decrease of 1.4% and a relative decrease of 13.4% of top deciles, with an OR of 0.853.

Of course, the aforementioned calculations are based on the assumption that the 'deciles' for calculated grades are expressed at the same standard as currently. Were the outcomes to be restandardised so that all deciles were equally represented, then of course at finals no noticeable difference in performance would be present, since of necessity 10% would remain in the top decile, etc. However, the 'academic backbone' would still be present, and overall poorer performance on statistically equated postgraduate exams[62].

## DISCUSSION

The present data make clear that under a half of predicted grades are accurate, with 45% being higher than attained grades, and 17% being lower. The data also show that attained grades are far better predictors of medical school performance than are predicted grades, which account for only about a third as much outcome variance as attained grades. Attained grades are also more reliable than predicted grades.

Validation is the bottom line for all measures used during selection, and in the present case, it is validation against assessment 5–8 years down the line from the original A levels, in both undergraduate and postgraduate assessments. That is strong support for what we have called 'the academic backbone', prior attainment providing the underpinning for later attainment, and hence there are correlations in performance at all stages of training from GCSEs through to medical degrees and on into postgraduate assessments.[5]

Our findings contradict suggestions that holistic judgements by teachers of predicted grades are better predictors of outcomes since teachers may know their students better than examiners. The immense efforts by exam boards and large numbers of trained markers to refine educational measurements is therefore gratifying and reassuring. Careful measurement does matter.

An important question is whether there is some variance in predicted and actual grades, which is complementary. We found that adding predicted grades to the model predicting outcomes improved the multiple correlation coefficient by only 0.05, accounting for only an additional 0.25% of variance. This suggests that predicted grades may provide a very small amount of additional information in predicting outcomes. What that information might be is unclear, and it is possible that it is what Petch called 'scholarly attitude'. At present though, it is worth remembering that *examination* grades at A-level are

primarily predicting further examination grades at the end of medical school, although EPM scores do include formal assessments of course work, and practical and clinical skills. If other outcome measures, perhaps to do with communication, caring or other non-cognitive skills were available, then predicted grades might show a greater predictive value.

The present data inevitably have some limitations. There is little likelihood of bias since complete population samples have been considered, and there is good statistical power with large sample sizes. Inevitably not all outcomes can be considered, mainly because the cohorts analysed have not yet progressed sufficiently through postgraduate training. However, those postgraduate outcomes which are included do show substantial effects which are highly significant statistically.

Our questions about predicted grades have been asked in the practical context of the cancellation of A-level assessments and their replacement by calculated grades, as a result of the COVID-19 pandemic. It seems reasonable to assume, given the literature on predicted grades, and particularly on forecasted grades, that calculated grades will probably have similar predictive ability to predicted grades, but perhaps will be a little more effective due to occurrence later in the academic cycle. Such a conclusion would be on firmer ground if exam boards had analysed the predictive validity of the data they had collected on forecasted grades, particularly in comparison with predicted and actual grades. Such data may exist, and if so, then they need to be seen. In their absence, the present data may be the best available guesstimates of the likely predictive validity of calculated rather than actual grades.

A potential limitation of our study is that we do not include the calculated and final grades for students who applied for admission in 2020; however, calculated and final grades for 2020 will be available in UKMED in 2021, and since that year group will also have the teacher-predicted grades submitted to UCAS, an immediate question of interest will be the extent of the correlation of the measures and hence whether teacher-predicted grades are indeed a proxy for calculated grades. Having said that, it will not be possible to calculate the predictive validity of teacher-predicted and calculated grades for a number of years until the cohort progresses through undergraduate training. Medium-term and long-term predictive validity inevitably take time to acquire, and practical decision-making sometimes has to be based on proxy and surrogate measures, with teacher-predicted grades at application to UCAS being a reasonable substitute. If it were the case that teacher-predicted grades for UCAS and teacher-estimated grades as a part of calculated grades were fundamentally discrepant, then serious questions would be raised about one or other set of estimates. The same applies to the teacher-estimated grades being used as a substitute for A levels in the summer of 2021, which will apply to the cohort applying for entry to medical school in 2021.

## Underprediction

Underprediction is a particular risk in cases where teachers do not know their students well or, in some cases perhaps, underestimate their ability because of attitude, personal characteristics or other factors. There is some evidence that teacher-assessed grades relate more to student personality than do grades in national examinations,[63 64] although effects were relatively weak. Any such biases are traditionally solved by the externality and objectivity of national examinations. Petch, once again, put it well, describing,

> instances, where, in the examination room, candidates have convinced the examiners that they are capable of more than their schools said that they were … Paradoxical as it will seem, examiners are not always on the side of authority; an able rebel can find his wider scope within the so-called cramping confines of an examination.[3] (p.29).

There is a clear echo here of the quote by Yasmin Hussein with which this paper began. Hussein's concerns are not alone, and the UKMACS study in April 2020 found concerns about fairness were particularly present in medical school applicants from non-selective schools, from black, Asian and minority ethnic applicants, from female applicants, and from those living in more deprived areas.[10]

### Effects of loss of schooling

A further consideration is more general and asks what the broader effects of the COVID-19 pandemic may be on medical education. Students at all levels of education have had teaching and learning disrupted, often extensively, and that is also true of all stages of medical education. The 2020 cohort of applicants/entrants will not have been assessed formally at A level. As well as meaning that they may only have calculated grades, which are likely to be less accurate, they also will have missed out on significant amounts of teaching. UK students who should have taken A-level exams in 2020 missed around 30–40 school days; those in the year below from whom 2021 medical school entrants will be drawn will have missed around 80 days. Burgess and Sievertsen,[65] using data from two studies,[66 67] estimate that 60 lost school days result in a reduction in performance of about 6% of an SD, which they say is, 'non-trivial' (and for comparison, a rule of thumb is that students in school improve by about one-third of an SD in each school year[68].) These effects are likely to differ also by socioeconomic background, particularly given variability in the effectiveness of home schooling. Applicants not taking A levels will also suffer from the loss of the enhanced learning that occurs when learners are tested—the 'testing effect'—for which meta-analyses have found effect sizes of about 0.50,[69 70] which is also non-trivial. Taken overall, 2020 entrants to medical school, and perhaps those in 2021 as well, may—without additional support—perform less well in the future as a result of missing out both on education and on its proper assessment.

## CONCLUSIONS

The events of 2020 as a result of the COVID-19 pandemic were extraordinary, and unprecedented situations occurred of which the cancellations of GCSE and A-level exam cancellations were but one example. The current study should not be seen as criticism of the response of Ofqual to that situation; given the circumstances in which it found itself, with examinations cancelled (when the Chair of Ofqual, Roger Taylor, had recommended socially distanced or delayed exams), Ofqual's solution to the problems had many obvious virtues. We began this paper by quoting a letter to a newspaper in March 2020 at the beginning of lockdown by a student taking GCSEs, and so it is probably appropriate to finish with a letter to a different newspaper by an A-level student. Written at the height of the A-level crisis, in August 2020, it raises many subtle, important and mostly neglected questions, ones which researchers will need to grapple with in the future:

> *Ofqual's* grading system appears to be lacking in advocates. Blinded by rhetoric about what protesters call a 'classist' algorithm, key facts have been overlooked. It is very clear that teachers are shockingly bad at predicting grades; using teacher predictions there will be a 12% inflation in higher grades compared with last year. While some centres predicted accurately, some centres predicted only the highest grades for their students. This U-turn from the government entails a huge injustice for the pupils who had fair and accurate predictions, as well as for those taking exams next year. In the zero-sum game of university applications, the results of these pupils make them appear weaker than they are. Irresponsible teachers who over-predicted their pupils' results ought to be ashamed that they too have thereby 'dashed the dreams' of many young people across the country. That it is less obvious does not make it any less true. (Letter to *The Times*, 19 August 2020, by Seb Bird, A-level student, Bristol).[71]

For most university applicants, there already existed predicted grades from the previous autumn when UCAS applications were submitted, but they would have been on average half a grade or so too high, being aspirational as much as realistic, and also for medical students would have been made by October 2019, whereas calculated grades would be based on teacher predictions in May 2020, although with several months of courses missing since March 2020.

In May 2020, we wrote that raw teacher-predicted grades would have wrecked much university planning, particularly coming so late in the year, after offers had been made, as numbers of acceptances would inevitably have been far too high.[7] That in fact happened, and quotas for university entries had to be abandoned

in August 2020, including for medicine, and that had knock-on effects into first-year university courses and probably beyond. There was also a risk that predicted grades could have been systematically higher from some schools than others—the ones with a tendency to call all of their 'geese' "swans"—and that probably applies also to the CAGs sent to examination boards and mostly eventually accepted without central standardisation in August 2020. The consequences of that will not become apparent for a few years.

This paper has provided evidence that the grades awarded to medical applicants in summer 2020 will probably not predict future outcomes with the same effectiveness as actual, attained grades, and that is a problem that universities and medical schools and postgraduate deaneries will have to work with, probably for many years as the 2020 cohort works through the system. It seems likely therefore, as Thomson has said, '… this year group will always be different…'.[2]

**Acknowledgements** We are grateful to Paul Garrud, Gill Wyness, Paul Newton, Colin Melville and Christian Woodward for their comments on earlier versions of this manuscript, and to Jon Dowell, Peter Tang, Rachel Greatrix and other members of the UKMED Research Group and Advisory Board for their assistance in fast-tracking the preprint of this paper, and for their comments on it. We also thank Tim Gill for providing us with an unpublished manuscript.

**Contributors** DTS prepared the data extracts, provided details on date sources and variable definitions where required and commented on manuscript drafts. ICM originated the idea for the study and discussed it with other authors throughout the project, and wrote the first draft of the manuscript. KW, DH, PAT, LWP, KYFC and DTS read, reviewed and commented on earlier drafts and contributed ideas, as well as approved the final draft, both of the preprint and of the present paper. ICM is the guarantor for the paper.

**Funding** KW is a National Institute for Health Research (NIHR) Career Development Fellow (NIHR CDF-2017-10-008) and is principal investigator for the UK Medical Applicants Cohort Studyand UKMEDP089 projects supported by the NIHR funding. DH is funded by NIHR grant CDF-2017-10-008 to KW. PAT's research time is supported by an NIHR Career Development Fellowship (CDF 2015-08-11), and PAT is also principal investigator for the UKMEDP051 project. LWP is partly supported by NIHR grant CDF 2015-08-11 to PAT, and a portion of his research time is funded by the UCAT board.

**Disclaimer** KW and DH state that this publication presents independent research funded by the National Institute for Health Research (NIHR). The views expressed are those of the authors and not necessarily those of the NHS, the NIHR or the Department of Health and Social Care. PAT states this research is supported by an NIHR Career Development Fellowship (CDF 2015-08-11). This paper presents independent research partly funded by the NIHR. The views expressed are those of the authors and not necessarily those of the NHS, the NIHR or the Department of Health and Social Care. KYFC is employed as the Head of Marking and Results at Cambridge Assessment English. The views expressed are those of the authors and do not represent the views of Cambridge Assessment. DTS is employed by the GMC as a data analyst working on the UK Medical Education Database(UKMED) project. The views expressed here are his views and not the views of the GMC. Data sources: UKMED, UKMEDP051 data extract generated on 13 May 2019. UKMEDP089 extract generated 21 January 2020. UKMEDP112 project, using UKMEDP051 and UKMEDP089 data, approved for publication on 29 May 2020. We are grateful to UKMED for the use of these data. However, UKMED bears no responsibility for their analysis or interpretation. UKMEDP051 data includes information derived from that collected by the Higher Education Statistics Agency Limited (HESA) and provided to the GMC (HESA Data). Source: HESA Student Record 2002/2003 to 2014/2015. Copyright Higher Education Statistics Agency Limited. The Higher Education Statistics Agency Limited makes no warranty as to the accuracy of the HESA Data, cannot accept responsibility for any inferences or conclusions derived by third parties from data or other information supplied by it.

UKMEDP051 and UKMEDP089 include Universities and Colleges Admissions Service (UCAS) data provided to the GMC (UCAS data). Source: UCAS (application cycles 2007 to 2018). Copyright UCAS. UCAS makes no warranty as to the accuracy of the UCAS Data and cannot accept responsibility for any inferences or conclusions derived by third parties from data or other information supplied by it. All data from HESA are required to be reported using their rounding and suppression criteria (https://www.hesa.ac.uk/about/regulation/data-protection/rounding-and-suppression-anonymise-statistics) and we have applied those criteria to all UKMED-based tables and values reported here.

**Competing interests** ICM is a member of the UKMED Research Group and the UKMED Advisory Board, and is also on the UK Medical Applicants Cohort Study advisory group. PAT is a member of the UKMED Research Group. PAT has previously received research funding from the ESRC, the EPSRC, the Department of Health for England, the UCAT Board and the GMC. In addition, PAT has previously performed consultancy work on behalf of his employing University for the UCAT Board and Work Psychology Group and has received travel and subsistence expenses for attendance at the UCAT Research Group. KYFC is a member of the UKMED Research Group and is an employee of Cambridge Assessment, a group of exam boards that owns and administers the BioMedical Admissions Test, UK GCSEs and A levels, and International GCSEs and A-levels. DTS is a member of the UKMED Research Group and the UKMED Advisory Board and is employed by the GMC as a data analyst working on the UKMED project.

**Patient consent for publication** Not applicable.

**Provenance and peer review** Not commissioned; externally peer reviewed.

**Data availability statement** Data are available upon reasonable request. Researchers wishing to re-analyse the data used for this study can apply for access to the same datasets via UKMED (www.ukmed.ac.uk).

**ORCID iDs**
I C McManus http://orcid.org/0000-0003-3510-4814
Katherine Woolf http://orcid.org/0000-0003-4915-0715
Lewis W Paton http://orcid.org/0000-0002-3328-5634
Daniel T Smith http://orcid.org/0000-0003-1215-5811

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

71  Bird S. *A-levels fiasco*. The Times, 19th August 2020.

