## [Reviewer comments · BMJ Open]

ARTICLE DETAILS

TITLE (PROVISIONAL)	The predictive validity of A-level grades and teacher-predicted grades in UK medical school applicants: A retrospective analysis of administrative data in a time of COVID-19
AUTHORS	McManus, Ian; Woolf, Katherine; Harrison, David; Tiffin, Paul; Paton, Lewis; Cheung, Kevin; Smith, Daniel

VERSION 1 – REVIEW

REVIEWER	Simmenroth, Anne University of Gottingen Center of Internal Medicine
REVIEW RETURNED	22-Jan-2021

GENERAL COMMENTS	Thank you for this relevant and interesting paper, surely this will be a worldwide problem in these days of pandemic. Abstract: the methods-section is missing. Introduction: Please add a few sentences about the way in which A-level grades are determined in the UK: what is anonymised normally? E.g. in some European countries the grades are given by the usual teachers and only oral examinations are observed by external auditors. The explanations are very detailed and specific for the UK-situation. International readers perhaps are more interested in general topics (correlation between A-levels and university-performance later also in other countries etc.). methods: please create a flow-chart which describes the way of the data resp. the time-lines of the different scores (predicted/attained/estimated/other grades) You mention figures in the results-section. I found only figures in the supplementary material. Because the entire text is very long, the reader needs figures also in the article. Please consider which figures/tables you are able to insert into the article and which paragraphs in the results-section you are able to delete. P 14. line 26-33: this would be better located in the discussion. The first half of the discussion is also interesting for non UK-readers because considerations about the pandemic, the cohort of "pandemic students," and the reasonableness of the application processes in general are addressed. I recommend to move the postscript to the appendix.
---

REVIEWER	Rimfeld, Kaili King's College London
REVIEW RETURNED	02-Mar-2021

GENERAL COMMENTS	This is an overall well written and methodologically sound paper and is definitely suited for publication in the BMJ Open. The topic is very important and of interest to educationalists, researchers and policy makers. The study uses a large sample and a
---

complete population data of applicants to medical school, thus is not likely to be affected by selection bias or attrition. It is, however, only generalisable to medical students (or applicants to medical school), an already highly selected sample. I have some major suggestions and some minor comments for the authors to consider.

While this is a very interesting piece of research, it is based on predicted grades and not the centre assessment grades (CAGs). The authors discuss these limitations but do not take them into account in their final conclusions. This is especially evident in the abstract that goes too far from the data. Predicted grades are given almost a year prior to actual exams, that is when students apply for universities. While it is reasonable to take predicted grades as a proxy for centre assessment grades, this is a proxy with limitations. We do not know what the correlation between teacher predictions and centre assessment grades is, or how much they were adjusted up or down (this adjustment could also be school-specific and based on their own algorithm). A full school year can make a great difference – this might be reflected in the study results as well. Did Ofqual set limits to CAGs? There is clearly more guidance given for teachers (e.g., <https://ofqual.blog.gov.uk/2020/05/15/making-grades-as-fair-as-they-can-be-advice-for-schools-and-colleges/>) and I would argue that we should trust our teachers to make these judgements (see <https://inews.co.uk/opinion/a-level-gcse-results-trust-teachers-exams-592499>).

I would also suggest that the literature review and expert opinions to be more balanced in the manuscript. There are many leading experts who argue that we should move away from the high-stake exam system (e.g. Sarah-Jayne Blakemore, or see <https://rethinkingassessment.com/rethinking-blogs/looking-to-medicine-for-the-cure-for-our-ailing-assessment-system/>).

The high correlations between teacher grades and exam scores in this manuscript support this; perhaps the teachers can assess the students quite well. I would argue that it is too early to conclude that holistic judgements of teachers are not better or at least equal predictors of outcomes, we are most certainly interested in other outcomes in life rather than exam performance alone. Using exams to predict exams will always be better than using teacher grades to predict exam grades, that is, whatever cognitive and non-cognitive predictors are involved in exam-taking explain individual differences in both assessment points.

EPM scores also involve clinical and practical work, in addition to exam performance, which is relevant to the mastering profession, as illustrated in this paper. It is clear that teachers are picking up something that exams do not, this is evident for the additional variance explained after accounting for actual grades in this study (given the limitations of these predicted grades given a year before and using predicted grades rather than using CAGs and albeit explaining tiny proportion of variance). Attained A-level grades predict EPM $r=0.297$ which is better than predicted grades $r=0.198$; however, one can argue that these predicted grades are similarly predictive, especially as these grades are given a year earlier. In general, though, these correlations are not great (compared to age-to-age correlations of educational achievement across compulsory education). Perhaps this points to a wider problem: maybe it is time to redesign the educational system and rely on selecting students to medical school based on more predictive measures. I suggest discussing this in the paper.

Figure 3 illustrates the correlation between predicted grades and exam performance. Actual GCSE exam results and actual A-level exam results correlate at .421, that is exam performance in a two-year interval. GCSE exam scores correlate with predicted A-level grade, a one-year interval here, at .452 and predicted grades with attained A-level grades .585, also a one-year interval – these are all reasonable correlations, exam-exam correlations are not higher than exam-teacher correlations and indicate that exam scores and teacher grades are assessing student performance equally well in this restricted sample of medical students.

The manuscript refers to exams as a gold standard measure of assessing achievement, even though it also later discusses the limitations of exam assessment. All assessments will inevitably include measurement error. High-stake exams are relying on a performance on a single day, students might be lucky as the authors eluded, but may also be affected by anxiety or illness. In addition, exams are often marked by a single marker, and the marker might be affected by tiredness, illness etc. Ofqual has reported about this issue before stating that 'more than one grade could well be a legitimate reflection of a student's performance' (see <https://www.gov.uk/government/news/response-to-sunday-times-story-about-a-level-grades>). Exam grades are often not graded with the same mark if gone through double marking (https://www.thetimes.co.uk/article/revealed-a-level-results-are-48-wrong-xsj33jvnh?--xx-meta=denied_for_visit%3D0%26visit_number%3D0%26visit_remaining%3D0%26visit_used%3D0&--xx-mvt-opted-out=false&--xx-uuid=c98a0a46eb271c4b822e6583ca022b28&ni-statuscode=acsaz-307;https://www.hepi.ac.uk/2019/02/25/1-school-exam-grade-in-4-is-wrong-thats-the-good-news/).

Perhaps it is time to start discussions on how to best evaluate students' ability and performance over the school years, and time to redesign the educational system. I am not suggesting testing children at all and are not suggesting that students' progress should not be monitored. For example, it is possible that in an increasingly technological society light-touch frequent testing can aid learning. Teachers are already doing an important job, this includes monitoring students' progress and making sure they understand key concepts. It is possible that far too much time goes on making sure students pass the exams with flying colours with the knowledge that this has important implications for the school league tables. If grades (either teacher-rated or exam grades) would not be of such high importance for both students and educators, then this would benefit both the more advantaged and also disadvantaged children. Maybe we should bring back coursework and test children more frequently, even if this is just once a term and use a modular final grade at the end of their A-level years.

It is great that medical schools already use other forms of selection such as U(K) CAT or BMAT and multiple mini interviews (MMIs), as explained in the manuscript. Perhaps this could carry more weight in the admission decisions? The manuscript states that U(K) CAT or BMAT and MMIs add little to the prediction of university outcomes, which is surprising. I suggest adding how well these assessments predict performance on their own.

It was really interesting to read about forecasted grades, which are more similar to CAGs. While overpredicting was more common than underpredicting, underpredicting was still substantial, but here again, I suggest not comparing the forecasted grades to actual exam grades without considering measurement error in exam marking. In addition, it is important to note that the authors consider absolute grades. Grade boundaries are not known for the teachers, and they change every year. An A* grade in one year could be an A in another year. These grade boundaries can explain over and under prediction/forecast by a grade, and in the majority of the cases, the grades do not differ by more than one grade.

In conclusion, I think this is a fantastic comprehensive study comparing several assessment methods and admissions/performance in medical school. I suggest rephrasing some of the conclusions, emphasising the limitations more and staying within study results since we really do not know how well CAGs in 2020 assessed students' academic achievement and ability.

Minor suggestions:

	1) Figures in the main manuscript are both tables and figures. 2) I suggest briefly explaining the results in the main text for all findings even if results are presented in SOM (e.g., EPQ and SQA Advanced Highers) 3) I do not think that the hypothetical experiment with computer error is needed in a scientific paper. The authors state that predictions are qualitatively similar to random allocation, but your data shows that the predictions and actual grades are highly correlated, and the predictions are over and underpredicting by one grade in general.
--	--

VERSION 1 – AUTHOR RESPONSE

Reviewer: 1

Prof. Anne Simmenroth, University of Gottingen Center of Internal Medicine

Comments to the Author:

REVIEWER. Thank you for this relevant and interesting paper, surely this will be a worldwide problem in these days of pandemic.

Response: Thanks for your comments; and indeed we think this could well be a problem in many countries.

REVIEWER. Abstract: the methods-section is missing. [Editor: this is OK - you have followed BMJ Open formatting guidelines]

Response: Thanks to the editor for that comment.

REVIEWER. Introduction: Please add a few sentences about the way in which A-level grades are determined in the UK: what is anonymised normally? E.g. in some European countries the grades are given by the usual teachers and only oral examinations are observed by external auditors.

Response: We have added a sentence saying that, “A-levels and SQA assessments, like other national examinations in the UK, are normally set, and marked anonymously, by examination boards which are entirely separate from schools, and teachers usually play no part in the assessment process”. There are occasional exceptions occur such as language examinations, where teachers act as interlocutors for a pre-arranged topic, with the exam recorded and marking then carried out externally. However most medical school applicants are not taking languages, and we have therefore omitted that comment from the main text. Additional brief clarificatory comments on the UK system have been placed throughout the manuscript – see the Track Changes version.

REVIEWER. The explanations are very detailed and specific for the UK-situation. International readers perhaps are more interested in general topics (correlation between A-levels and university-performance later also in other countries etc.).

Response: We agree that the analyses and the description are very UK-centric, but this is a research paper specifically looking at the UK experience, and not a general review of international practice. If reading papers about the German, Austrian or French experience we would expect to read a similar level of detail for those papers, in order to interpret them properly. Our experience is that all national examination systems are idiosyncratic in their own ways (and while we could attempt to talk about, say, the Abitur in Germany or the Baccalaureate in France, or other assessments, we feel that would not add value to a study focussing on medical selection in the UK). We have added some references to the introduction to indicate that A-levels are known to correlate well with performance in medical school, which we hope is what international readers would wish to know. “A-levels are good predictors of performance at university in general⁴, and at medical schools specifically^{5 6}.”

REVIEWER. Methods: please create a flow-chart which describes the way of the data resp. the time-lines of the different scores (predicted/attained/estimated/other grades)

Response: Thank you for this good idea. We have created an annotated lby chart, which we have used before to indicate the structure of the rather complex UKMED data, and the timing of the various measures. Hopefully it gives an idea of the flow of typical students through the system in different cohorts, the timing of the various measures, and the availability of data. But these are only typical

students – there are many variants on the theme. We have included an extended description of figure 3 at the point where it is first mentioned.

REVIEWER. You mention figures in the results-section. I found only figures in the supplementary material. Because the entire text is very long, the reader needs figures also in the article. Please consider which figures/tables you are able to insert into the article and which paragraphs in the results-section you are able to delete. [Editor: this could make the article more readable, particularly for non-native English speakers]

Response: We followed the usual convention of placing figures at the end of the main paper (and the BMJ Open submission portal does that automatically). We presume that when the paper is edited that the sub-editors will incorporate figures into the main text. Figures 1 to 5 of the main paper were included at pages 34 to 38 of the proof that we were sent by BMJ Open, and we are unclear why they were not present in the reviewer's version of the paper, but it was a long and complex PDF file. In answer to the specific question from the editor, we assumed that all figures included in the main paper (i.e. on pages 34 to 38 of the original proof, and now figures 1 to 6) will be included in the main paper. The figures are, in our view, integral to the main paper.

REVIEWER. "Please consider ... which paragraphs in the results-section you are able to delete"

Response: This is a difficult question to answer. Neither the reviewer nor the editor mentions any particular paragraphs which seem ripe for deletion, and neither is there any clear word limit which we may need to meet. It is possible that some may regard some of the discussion as being outdated, but that in part as it was drafted in May 2020 before A-level results were announced (and the medRxiv preprint of June 2020 makes that clear). The paper as submitted here was written in the autumn of 2020, and submitted in November 2020, at which time no decisions had been made about the handling of A-levels for 2021. Decisions on that were made early in 2021, while this paper was but a third of its way through the six months spent in review at BMJ Open, and only now have revisions been requested. What is now seen as perhaps meriting deletion is very unclear. The paper is both a historical document and a document describing a process that is still in progress for results in the summer of 2021 (and A-levels results will be announced on August 10th 2021). We could try and update the discussion but that could involve us continually chasing a moving target. We think a clear steer is required at this point from the Editor, but hope that the current draft is a reasonable compromise, and that a fast decision can be made without further extensive revisions being required. In the meantime we have moved the Postscript (see below), and removed the sections marked for schools and applicants, referring the reader to the pre-print. We have also made numerous minor changes throughout the manuscript (see the Track Changes version) in response to the long delay before revision began, which we hope clarifies and situates the paper.

REVIEWER. P 14. line 26-33: this would be better located in the discussion.

Response: The lines read, "Correlations between the measures are summarised in Figure 3. Large differences in Ns reflect some measures being used in applicants during selection, and others being outcome measures that are only present in entrants, as well as the smaller numbers of doctors who had progressed to postgraduate assessments. The distinction is emphasised by dividing the correlation matrix into three separate parts. Correlations of selection and outcome measures necessarily show range restriction because candidates have been selected on the basis of the selection measures, and likewise doctors taking postgraduate examinations may be self-selected for earlier examination performance." Of the four sentences, the first three describe the results in figure 3, and presumably need to be in the results section. The comments about range restriction could be *in the Discussion*, but surely the sentence is needed where the reader encounters the correlation matrix, which is in the Results section. If the Editor feels that this sentence needs to be moved we will of course do it. However the section now includes formal measures of range restriction making it inappropriate for the discussion.

For the effects of range restriction we have also now included the construct-level predictive validities, which require calculation of range restriction (the parameter uX), and have included CLPVs and uX , with description, in figure 3. We hope that clarifies things, and also describes an important part of the interpretation of data such as these.

REVIEWER. The first half of the discussion is also interesting for non UK-readers because considerations about the pandemic, the cohort of "pandemic students," and the reasonableness of the application processes in general are addressed.

Response: We are glad that the comments may be helpful more broadly than in the UK.

REVIEWER. I recommend to move the postscript to the appendix.

Response. We have followed the reviewer's suggestion and moved the postscript to the supplementary information.

Reviewer: 2

Dr. Kaili Rimfeld, King's College London

Comments to the Author:

REVIEWER. This is an overall well written and methodologically sound paper and is definitely suited for publication in the BMJ Open. The topic is very important and of interest to educationalists, researchers and policy makers. The study uses a large sample and a complete population data of applicants to medical school, thus is not likely to be affected by selection bias or attrition. It is, however, only generalisable to medical students (or applicants to medical school), an already highly selected sample. I have some major suggestions and some minor comments for the authors to consider.

Response: Thank you for those positive comments on the methodology and the suitability for publication. We agree that the study is only applicable to medical students and medical school applicants, but that is the population which medical education studies, and BMJ Open specifically publishes material research relevant to medicine and medical practice. We would like to hope that similar analyses may be possible by other researchers for university entrants in general.

On the question of generalisation, all university applicants have teacher-predicted grades provided to UCAS, and in the context of the events of 2020, all university applicants selected from applicants with centre-assessed grades, and their entrants came from that group. Medical students do have high A-level grades, but there still seems reason to believe that the same processes are relevant to all university applications where selection occurs.

REVIEWER. While this is a very interesting piece of research, it is based on predicted grades and not the centre assessment grades (CAGs). The authors discuss these limitations but do not take them into account in their final conclusions. This is especially evident in the abstract that goes too far from the data. Predicted grades are given almost a year prior to actual exams, that is when students apply for universities. While it is reasonable to take predicted grades as a proxy for centre assessment grades, this is a proxy with limitations. We do not know what the correlation between teacher predictions and centre assessment grades is, or how much they were adjusted up or down (this adjustment could also be school-specific and based on their own algorithm). A full school year can make a great difference this might be reflected in the study results as well. Did Ofqual set limits to CAGs? There is clearly more guidance given for teachers (e.g., <https://ofqual.blog.gov.uk/2020/05/15/making-grades-as-fair-as-they-can-be-advice-for-schools-and-colleges/>) and I would argue that we should trust our teachers to make these judgements (see <https://rethinkingassessment.com/rethinking-blogs/looking-to-medicine-for-the-cure-for-our-ailing-assessment-system/>).

Response: "the abstract ... goes too far from the data." The abstract has been extensively revised (see Track Changes) and now is careful to make clear that teacher-estimated grades are a "plausible proxy" for calculated grades. The title has also been changed to include the phrase

“teacher-predicted grades” which we now make clear is a phrase we use only to refer to grades submitted to UCAS at the time of application by applicants.

We make clear in the manuscript that our analysis is entirely of grades predicted by teachers at the time of the application to UCAS. At the time this paper was written the CAGs were not available to researchers, and therefore the teacher-assessed grades were used as a proxy (and one of great interest in their own right anyway since they are routinely used in university selection processes, and it is surprising that there have been no previous studies of their predictive validity). There are inevitably limitations in the use of a proxy, but neither we nor the reviewer can have any objective sense of those limitations. We discuss the complex issues in the paper, and it is our belief that grades predicted for UCAS are indeed a reasonable proxy. Even if they are not, it is still clear that the teacher-predicted grades for UCAS applications have much lower predictive validity than do actual A-level grades which has implications for student selection by universities. The CAGs themselves should be available to medical education researchers towards the end of 2021, and then further questions will be answerable. In the interim it makes sense to consider the proxies and determine their suitability when the data become available.